# Neutrophil-avid nanocarrier uptake by STAT3 dominant-negative hyper-IgE syndrome patient neutrophils

Kathryn M Rubey[1], Alexandra Freeman[2], Alexander R Mukhitov[3], Andrew J Paris[3], Susan M Lin[3], Ryan Rue[3], Hossein Fazelinia[4], Lynn A Spruce[4], Jennifer Roof[4], Jacob S Brenner[3,5], Jennifer Heimall[1,*], Vera P Krymskaya[3,*]

**Recurrent infections are a hallmark of STAT3 dominant-negative hyper-IgE syndrome (STAT3 HIES), a rare immunodeficiency syndrome previously known as Jobs syndrome, along with elevated IgE levels and impaired neutrophil function. We have been developing nanoparticles with neutrophil trophism that home to the sites of infection via these first-responder leukocytes, named neutrophil-avid nanocarriers (NANs). Here, we demonstrate that human neutrophils can phagocytose nanogels (NGs), a type of NAN, with enhanced uptake after particle serum opsonization, comparing neutrophils from healthy individuals to those with STAT3 HIES, where both groups exhibit NG uptake; however, the patient group showed reduced phagocytosis efficiency with serum-opsonized NANs. Proteomic analysis of NG protein corona revealed complement components, particularly C3, as predominant in both groups. Difference between groups includes STAT3 HIES samples with higher neutrophil protein and lower acute-phase protein expression. The study suggests that despite neutrophil dysfunction in STAT3 HIES, NANs have potential for directed delivery of cargo therapeutics to improve neutrophil infection clearance.**

## Introduction

Jobs syndrome is a rare form of immunodeficiency associated with a classic clinical triad of elevated IgE levels, recurrent skin and lung infections, and eczema. *Staphylococcus aureus* is the main cause of skin infections in these patients, and the initial recurrent sino-pulmonary infections. Pneumonias then characteristically lead to bronchiectasis and cavitary lung lesions with severe secondary fungal infections with *Aspergillus* (Holland et al, 2007; Heimall et al, 2010; Gernez et al, 2018; Zhang et al, 2023). Another hallmark infectious presentation is mucocutaneous candidiasis. The molecular basis of this clinical presentation was first demonstrated to be associated with autosomal dominant-negative mutations in the signal transducer and activator of transcription 3 gene (*STAT3*), leading to impaired STAT3 signaling, as Job Syndrome has been redefined as STAT3 dominant-negative hyper-IgE syndrome (STAT3 HIES). STAT-JAK signaling is a critical intracellular pathway for activation of target genes after cytokine binding to cell surface markers. STAT3 is particularly critical to the function of the pro-inflammatory cytokine interleukin 6 (IL-6). In turn, IL-6 has several key immune functions including providing a stimulus for neutrophil migration (Fielding et al, 2008; Milner et al, 2008). The recurrent infections seen in patients with STAT3 HIES can be attributed in part to dysfunctional neutrophils, with impaired chemotaxis and decreased bacterial clearance (Mintz et al, 2010; Farmand et al, 2018). This patient population is typically treated with many courses of systemic antibacterial and antifungal therapies over their lifetime, which can lead to organ toxicities and side effects. Therefore, the exploration of alternative methods for drug delivery may decrease the morbidity for patients with STAT3 HIES.

Nanosized drug carriers are an emerging therapeutic area for directed delivery of pharmaceuticals. We have previously described a nanocarrier class with tropism to neutrophils, "neutrophil-avid nanocarriers" (NANs) (Myerson et al, 2022; Rubey et al, 2022). These nanoparticles, of the prototypic particle being a lysozyme-dextran nanogel (NG), are avidly taken up by neutrophils, especially marginated neutrophils residing in the lumen of the pulmonary capillaries under inflammatory states. This suggests that the NANs could be used in pathologies such as pneumonia, COVID-19, and ARDS. We have previously shown that the neutrophil tropism of these particles is increased when serum opsonins bind the particles (Myerson et al, 2022). Although NGs are the prototypical NAN, our lab has been developing a diverse class of nanoparticles to target immune cells. Given the known neutrophil defects in STAT3 HIES patients, this population is of particular interest for neutrophil-specific drug delivery. This study uses lysozyme-dextran nanogels (hereafter referred to as "nanogels" or NGs) which are one

[1]Department of Pediatrics, Children's Hospital of Philadelphia, Philadelphia, PA, USA  [2]Laboratory of Clinical Immunology and Microbiology, National Institute of Allergy and Infectious Diseases, National Institutes of Health, Bethesda, MD, USA  [3]Department of Medicine, University of Pennsylvania, Philadelphia, PA, USA  [4]The Proteomics Core Facility, The Children's Hospital of Philadelphia, Research Institute, Philadelphia, PA, USA  [5]Department of Pharmacology, University of Pennsylvania, Philadelphia, PA, USA

Correspondence: rubeyk@chop.edu
*Jennifer Heimall and Vera P Krymskaya are co-senior authors

of our diverse classes of untargeted nanoparticles with neutrophil-tropism, known as NAPs. In this present study, we have chosen to use NGs as our prototypical particle, given that its components are safe for human use, its prolonged nanoparticle shelf-life, and potential for a very high drug-to-carrier mass ratio (Myerson et al, 2018, 2019). These drug carriers could be beneficial to patients with STAT3 HIES by delivering cargo to their neutrophils to improve their response to these patients' high burden of bacterial and fungal pneumonia.

In this feasibility study, we first investigated if human neutrophils are able to phagocytose NGs and whether the serum opsonins are critical to this functionality. Then, we investigated for differences in NG phagocytosis between neutrophils from healthy controls compared with those from patients with STAT3 HIES. In addition, we assessed if patient serum is able to effectively opsonize NGs to increase their uptake by neutrophils.

## Results and Discussion

In this study, we compared the NG phagocytosis activity of neutrophils from six patients with STAT3 HIES to five healthy controls (Table 1). As a first investigative approach, we demonstrated that our previous murine findings are translatable to human neutrophils in two aspects (Myerson et al, 2022): first, that NGs are phagocytosed by human neutrophils and particle uptake is enhanced by serum opsonization of particles (Fig 1); second, that complement component proteins are the primary opsonin of nanogels incubated in human serum (Fig 2).

Patients with STAT3 HIES are known to have dysfunctional neutrophils. To determine if NAPs have the potential to be a viable drug delivery platform for STAT3 HIES patients, our first tests were to compare neutrophil phagocytosis of NGs from healthy controls to those from patients with STAT3 HIES using flow cytometry (Fig 1A). For naive, unopsonized NGs, both healthy donor and STAT3 HIES patient neutrophils were able to take up NGs with no statistically significant difference in the number of neutrophils that were positive for NG fluorescence or the mean fluorescence of neutrophils measured after incubation (Fig 1B and C). For the opsonized NGs, both sets of neutrophils showed an increase in NG uptake after serum opsonization of the particles, similar to previous murine studies. However, although there is a similar increase in the percent of neutrophils positive for some NG uptake, the mean fluorescence of the neutrophils is lower in the STAT3 HIES patient neutrophils. We believe this represents that, as expected, STAT3 HIES patient neutrophils have an increased affinity for opsonized NGs; however, they are unable to increase their phagocytosis of opsonized NGs by the same magnitude as healthy donor neutrophils (Fig 1B). This is again seen in the representative histogram shown in Fig 1C, a quantification of measured NG fluorescence. The lines for the naive NG are overlying each other and seen with a peak around $10^3$. There is a peak shift to the right in both serum-opsonized NG conditions, showing an almost 100-fold increase in measured fluorescence; however, the magnitude of the peak shift is less in the STAT3 HIES neutrophils when compared with that seen from healthy donor neutrophils. Our previous work has shown that murine neutrophils are able to phagocytose and co-localize NGs with bacterial particles

(Rubey et al, 2022). To determine if NGs remain membrane-bound or are phagocytosed by human neutrophils, we used confocal microscopy. Through z-stack analysis, we were able to confirm that the NGs are internalized by healthy donor human neutrophils, rather than remaining membrane-bound as shown in the representative image in Fig 1D and E.

Our previous work has identified a predominance of complement components, specifically complement component 3 or C3, in the NG protein corona after serum opsonization. We have in addition shown that complement opsonization is necessary for the increased phagocytosis of NGs seen after serum exposure (Myerson et al, 2022). Little, however, is known about the protein corona on NG particles from patients with STAT3 HIES. Based on the increased phagocytosis of NGs after opsonization seen in Fig 1 in both healthy control samples and those derived from patients with STAT3 HIES, we determine whether the patient serum is able to create a similar protein corona on NGs with a predominance of complement components binding. To investigate this, we used mass spectrometry for proteomic analysis. The protein analysis revealed 555 overlapping peptides in the NG corona formed from both healthy donor and STAT3 HIES patient serums. In addition, there were 48 unique peptides from healthy donor serum samples and 428 unique peptides from the patient serum samples (Fig 2A). When comparing the top 10 peptides identified bound to NGs, we again found a predominance of complement components. When looking at the collective groups, both healthy donor serum exposed NGs and STAT3 HIES patient serum exposed NGs had C3 as the most abundantly bound protein, with C4 and C5 also in the top 10. NGs from both serum samples also showed albumin, the most abundant serum protein, as the second most bound protein on particles (Fig 2B). The remainder of the top 10 identified peptides from the NG protein corona shows some differences. On healthy donor serum-opsonized NGs, we see fibronectin, an extracellular matrix protein, and apolipoprotein As, a component of HDL. In comparison, STAT3 HIES patient serum-opsonized NGs round out their top 10 bound peptides with an apolipoprotein B, a component of LDL, and the cytoskeletal proteins talin-1, alpha-actinin-1, and myosin-9 (Fig 2B). When looking at the individual spectral counts of the C3 and C5 measured in the samples, we found that although there was no statistically significant difference in the amount of C3 and C5 measured between the groups, there is an apparent trend in attenuated C3 and C5 levels in STAT3 HIES patient serum-opsonized NGs (Fig 2C and D).

Proteomic results were in addition analyzed for the peptides with statistically significant differences between the healthy donor and STAT3 HIES patient samples. We included in our results the identified peptides that had both a difference between the healthy donor and patient groups and were present in at least 50% of the samples of either group. We found that these peptides fell into four broad categories: neutrophil proteins, cytoskeletal proteins, acute-phase proteins (APP), exosome and platelet proteins (Table 2 and Fig 3). We see that for neutrophil, cytoskeletal, and exosome proteins, there is an increase in the proteins measured on the NGs opsonized with STAT3 HIES patient serum. Conversely, there is a decrease in the APPs bound to NGs after opsonization with STAT3 HIES patient serum (Table 2); this likely reflects the critical role of STAT3 in stimulation of APP production.

**Table 1.  Patient demographics and clinical information.**

| Sample | Age | Sex | Race | Patient clinical information |
|---|---|---|---|---|
| Healthy 1 | 29 | M | Caucasian | |
| Healthy 2 | 38 | M | Caucasian | |
| Healthy 3 | 60 | F | Caucasian | |
| Healthy 4 | 33 | M | Caucasian | |
| Healthy 5 | 42 | M | Caucasian | |
| Patient 1 | 18 | M | Hispanic | AD STAT3 HIES (c.1144C>T) diagnosed at 7 y/o with, at that time, IgE 1,850 IU/ml; AEC 606 cells/µl; and a clinical history of pneumonia, two episodes of MRSA boils of the skin, recurrent fungal fingernail infections, mild eczema, two retained primary teeth, and three episodes of minimal trauma fractures. He now has recurrent skin abscesses, bronchiectasis, and inflammatory bowel disease. |
| Patient 2 | 26 | F | African American, non-Hispanic | AD STAT3 HIES (genetic testing report not available) diagnosed based on clinical symptoms at 2 yr of age which included recurrent MSSA skin abscesses, IgE 3,970 IUI/ml, AEC 664 cells/µl with confirmatory genetic testing performed at age 12 years. She now has difficulty with recurrent polymicrobial skin abscesses, particularly of the breast, requiring hospitalization and surgical drainage; recurrent vaginal candidiasis; mild bronchiectasis with small pneumatoceles; and eczema which is controlled with dupilumab. |
| Patient 3 | 27 | F | Hispanic | AD STAT3 HIES (c. 1145 G>A) diagnosed at 2 y/o. Her highest recorded IgE is 36,307 IU/dl, and highest recorded eosinophil count 330 cells/µl. Clinical infectious history of recurrent skin abscesses, eczema, mucocutaneous candidiasis, recurrent sinopulmonary infections with pneumatocele formation, and an episode of bacterial sepsis. There is also noninfectious history of retained primary teeth, mild scoliosis, and recurrent minimal trauma fractures. |
| Patient 4 | 42 | F | Caucasian | AD STAT3 HIES (c. 1144 C-T) diagnosed at 21 y/o. Her highest recorded IgE is 3,880 IU/dl, and highest recorded eosinophil count 1,440 cells/µl. Clinical infectious history of recurrent skin abscesses, newborn rash, eczema, mucocutaneous candidiasis, shingles as a teenager after natural infection, recurrent sinopulmonary infections with pneumatocele formation, noninfectious history of retained primary teeth and mild scoliosis but no fractures, and history of lymphoma. |
| Patient 5 | 7 | M | African American | 7 y/o African American male with AD STAT3 HIES (c.1144C>T) diagnosed at one y/o. His highest recorded IgE is 9,020 IU/dl, and highest recorded eosinophil count 1,500 cells/µl. Clinical infectious history of recurrent skin abscesses with both MRSA and MSSA, mastoidosis with subdural empyema, and pneumonia. There is also noninfectious history of newborn rash and eczema. |
| Patient 6 | 60 | M | Caucasian | 60 y/o White male with AD STAT3 HIES (c.1145G>A) diagnosed at 7 y/o. His highest recorded IgE is 1,040 IU/dl, and highest recorded eosinophil count 1,768 cells/µl. Clinical infectious history of recurrent skin abscesses with both MRSA and MSSA, recurrent sinusitis and pneumonia with pulmonary aspergillosis and mucocutaneous candidiasis, a *Bordetella bronchiseptica* lung infection, and bursitis. He also experienced early shingles at the age of 44 yr. There is also noninfectious history of eczema, severe scoliosis, recurrent minimal trauma fractures, typical associated facial features, and a high arched palate. From a pulmonary perspective, he also has experienced bronchiectasis with pneumatocele formation and metastatic squamous cell lung cancer. Other complications included a myocardial infarction associated with a coronary artery aneurysm, a GI bleed associated with aneurysm of the mesenteric artery, and severe degenerative cervical spine disease requiring surgical stabilization. |

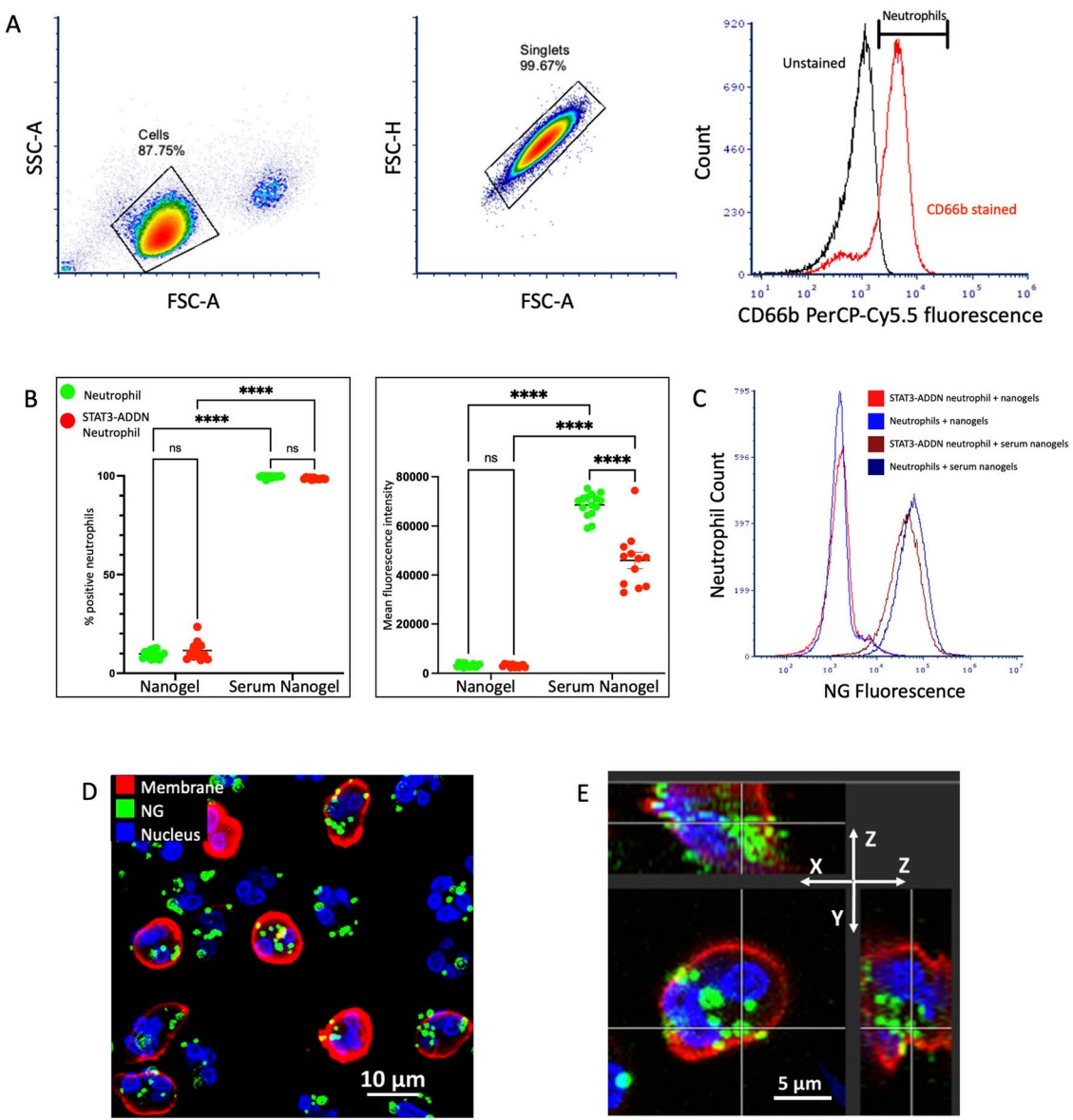

**Figure 1. Serum opsonization of fluorescent nanoparticles enhance their phagocytosis by human neutrophils.**
**(A)** Neutrophil isolation was performed on whole blood samples. Gating strategy for identifying CD66b-positive neutrophils for analysis is shown. **(B)** Neutrophils were isolated from whole blood from healthy donors and STAT3 HIES patients. NGs were incubated for 60 min in either media or donor-matched serum. Neutrophils were then exposed to the FITC-NGs, and neutrophil uptake was quantified with flow cytometry. In the first panel, we present the aggregated data from six patients and five controls with the percentage of present neutrophils that were positive for fluorescence, and therefore, NG uptake is seen. Similar to our previous mouse work, we found that there were more neutrophils positive for NG fluorescence when the particles are serum-opsonized before neutrophil exposure. There was no significant difference in the percentage of neutrophils with NG between the two groups. In the second panel, mean fluorescence intensity is used as a measure of the amount of NG uptake by the neutrophils. We again found that human neutrophils react to NGs similar to our previous mouse work, with more NG fluorescence seen after NG serum opsonization. This suggests that not only are more neutrophils taking up particles but also there are more particles per neutrophil after serum opsonization. Unlike in the first panel, there are differences seen between the healthy donor neutrophils and the STAT3 HIES donor neutrophils in the serum NG group. The mean fluorescence intensity in STAT3 HIES neutrophils is significantly lower than that seen in the healthy neutrophils, suggesting that although neutrophils ingest NGs after opsonization, the STAT3 HIES neutrophils are not able to uptake as many particles per neutrophil. **(C)** An example set of the raw flow cytometry data for one patient and one control. Events shown are live, singlet cells leukocytes positive for CD66b staining. The flow cytometry histograms show the same fluorescence peak when looking at the NG condition (royal blue and red lines); however, in the serum NG (maroon and navy lines), there is a peak shift. The healthy donor neutrophil has a shift to the right, corresponding to more fluorescence and therefore more particle uptake by the neutrophils. **(D)** Confocal microscopy was use to evaluate the cellular location of NGs. Healthy donor neutrophils were isolated from whole blood and incubated with serum-opsonized FITC-NGs. DAPI was used to for nucleus stain (blue). The membrane was stained with red DiD. **(E)** Orthogonal projection from z-stack analysis; NGs are seen within the neutrophil, rather than associated with the membrane, suggesting that they are phagocytosed and internalized by the neutrophils.

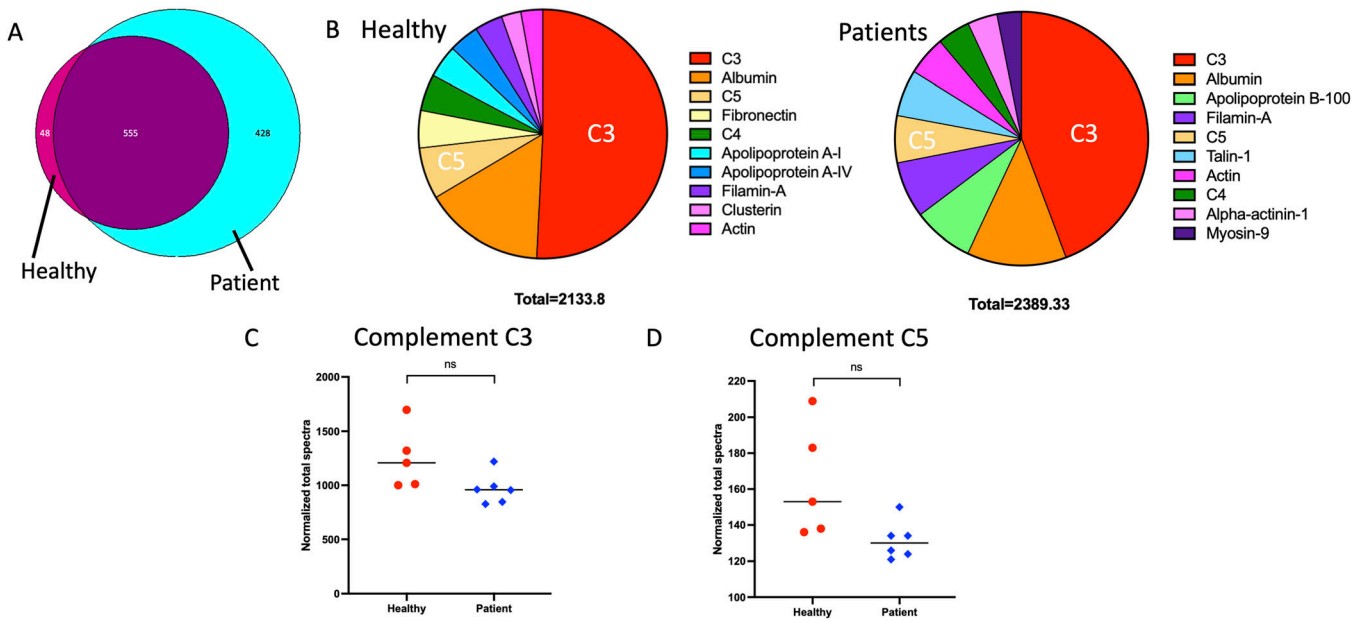

**Figure 2. Serum opsonization results in a complement-predominated protein corona on the NG particles.**
NGs were incubated in either healthy donor or STAT3 HIES patient serum. They were then washed three times and prepared for mass spectrometry and protein profiling. **(A)** The protein signature of the NG coronas were compared and revealed 555 overlapping peptides (purple) found in the NGs prepared from both healthy and patient serum. There were 48 unique peptides identified on the healthy donor serum samples (pink), and 428 unique peptides from the patient serum samples (aqua). **(B)** Comparison of the 10 most abundant peptides in the protein corona of the healthy donor serum NG on left and the STAT3 HIES patient serum samples on the right. Both serums bound a predominance of C3, a complement cascade component, on the NG particles. In addition, they both have C5 and C4, other complement cascade components, identified in the top 10 peptides per sample. Albumin, the most abundant serum protein, is the second most bound peptide in both healthy donor and patient serum exposed NGs. Although the remainder of the top 10 peptides identified have a few overlapping proteins, such as actin and filamin-A, the remainder of the highest abundance peptides in the protein corona from the STAT3 HIES patient were from cytoskeletal peptides. **(C, D)** The normalized total spectra counts for C3 and C5, respectively, of the individual healthy and patient samples. The median counts of the groups are not statistically significantly different when compared.

It has been previously reported that the neutrophil dysfunction seen in patients with STAT3 HIES is in the chemotaxis of the cells (Hill et al, 1974; Hashemi et al, 2017). We found that many of the proteins identified in higher amount bound to NGs are associated with phagocytosis, such as Rac family small GTPase 2 (RAC2) and integrin subunit alpha M (ITGAM); cell adhesion, such as FERM domain containing kindlin 3 (FERMT3), integrin subunit beta 2 (ITGB2), and ITGAM; and bactericidal activities, such as cathepsin G (CTSG), cytochrome b-245 beta chain (CYBB), neutrophil elastase (ELANE), and bactericidal permeability increasing protein (BPI) (Table 2, Figs 3A and S1A and B). We hypothesize that the increase in proteins necessary for other neutrophil functions may be a compensatory response. It is possible that because there are very few neutrophils that are able to home to the site of infection or inflammation, continued signaling causes increased activation of other functional neutrophil proteins, such as the ones needed for phagocytosis. Of note, these samples also had an increase in serpin family B member 1 (SERPINB1), which is a proteinase inhibitor necessary to maintain homeostasis of the neutrophil proteinases such as elastase, cathepsin G, and proteinase-3, all of which were seen in high numbers on the STAT3 HIES patient serum-opsonized NGs. The presence of these proteins in higher amounts in STAT3 HIES-opsonized NGs may be related to the neutrophil dysfunctional chemotaxis. In addition to the loss of STAT3's negative regulation of neutrophil function, we hypothesize that the STAT3 HIES patient neutrophils are trying to work, but because some functions are

ineffective, there is a compensatory up-regulation in the functional proteins, leading to the results seen here.

STAT3 is an important activator of many cytokines, including IL-6, an interleukin essential to the acute phase response signaling cascade. In response to injury or infection, local inflammatory immune cells secrete cytokines which will stimulate the liver to produce APPs via its major mediator, IL-6 signaling (Alonzi et al, 2001; Tanaka et al, 2014; Robinson et al, 2016; Zhou et al, 2016). With impaired STAT3 signaling, STAT3 HIES patients have dampened acute phase responses, and congruent with that, we found decreased APPs on the patient opsonized NGs. Although there is sample to sample variability (Figs 3C and S1A and B), we found when comparing the groups, fourfold less fibrinogen gamma chain (FGG) and fibrinogen beta chain (FGB) and twofold less fibrinogen alpha chain (FGA), extracellular matrix protein 1 (ECM1), and fibronectin 1 (FN1) on patient serum-opsonized NGs when compared with the healthy serum-opsonized NGs (Table 2).

In Figs 3D and S1A and B, we see that there were increased amounts of platelet and exosome signature proteins in the patient serum-opsonized NGs as compared with the healthy donor serum opsonization. The neutrophil-platelet interaction has been described as a key component of the inflammatory response. Activated platelets and platelet exosomes can initiate and amplify various neutrophil responses, like phagocytosis and NETosis (Pervushina et al, 2004; von Hundelshausen et al, 2005; Hurley et al,

**Table 2. Identified proteins of interest with the log-fold change of protein in STAT3 HIES patients compared with the healthy controls.**

| Protein | Log-fold change when compared with healthy donor samples | P-value |
|---|---|---|
| **Neutrophil proteins** | | |
| RAC2 | 5.197 | $8.60 \times 10^{-4}$ |
| CTSG | 4.768 | 0.004 |
| CYBB | 3.537 | 0.005 |
| FERMT3 | 3.493 | 0.007 |
| SERPINB1 | 3.124 | 0.018 |
| ITGB2 | 3.519 | 0.020 |
| ELANE | 2.877 | 0.025 |
| GCA | 4.417 | 0.026 |
| BPI | 3.103 | 0.029 |
| ITGAM | 3.391 | 0.043 |
| FCGR3B | 2.916 | 0.045 |
| **Cytoskeletal proteins** | | |
| VASP | 4.411 | $2.03 \times 10^{-5}$ |
| MYH9 | 4.831 | 0.002 |
| MYL12A | 3.559 | 0.004 |
| ARPC4-TTLL3 | 3.564 | 0.004 |
| ARPC3 | 3.086 | 0.006 |
| **Exosome and platelet proteins** | | |
| GP1BB | 5.388 | $3.63 \times 10^{-4}$ |
| GP1BA | 4.163 | $7.11 \times 10^{-4}$ |
| CD63 | 4.192 | 0.090 |
| PECAM1 | 2.630 | 0.090 |
| CD47 | 2.978 | 0.021 |
| CD9 | 2.023 | 0.027 |
| STX11 | 2.012 | 0.026 |
| ITGA2B | 2.159 | 0.043 |
| ITGB3 | 2.300 | 0.050 |
| **Acute-phase proteins** | | |
| FGG | −4.649 | $3.69 \times 10^{-5}$ |
| FGB | −4.281 | $1.04 \times 10^{-4}$ |
| FGA | −2.447 | 0.002 |
| ECM1 | −2.748 | 0.005 |
| FN1 | −2.767 | 0.006 |

2015; Wienkamp et al, 2022). Platelets also promote recruitment of neutrophils into inflamed areas (Zarbock et al, 2007). It has previously been reported that STAT3 HIES patients have increased NETs (Goel et al, 2021). We hypothesize that the increased platelet and exosome proteins found in the protein corona of NGs opsonized with patient serum as compared with healthy donor serum may be tied to the patient's dysfunctional neutrophils. The platelets may be up-regulated and release more exosomes in patients with impaired neutrophil chemotaxis in an effort to recruit more neutrophils to the sight of infection and inflammation and to up-regulate their NETosis of present neutrophils. We believe the increased cytoskeletal proteins seen in patient samples (Fig 3B) may be related to our platelet findings. Cytoskeletal elements are found in both platelet exosomes and neutrophil NETs (Fisk et al, 2006; Vorobjeva & Chernyak, 2020). Based on our other findings, we propose that the increase in cytoskeletal proteins is because of our results suggesting both platelet exosomes and NETosis are increased in patients.

This study has several limitations. First, all experiments performed were in vitro studies, and the process of neutrophil isolation may

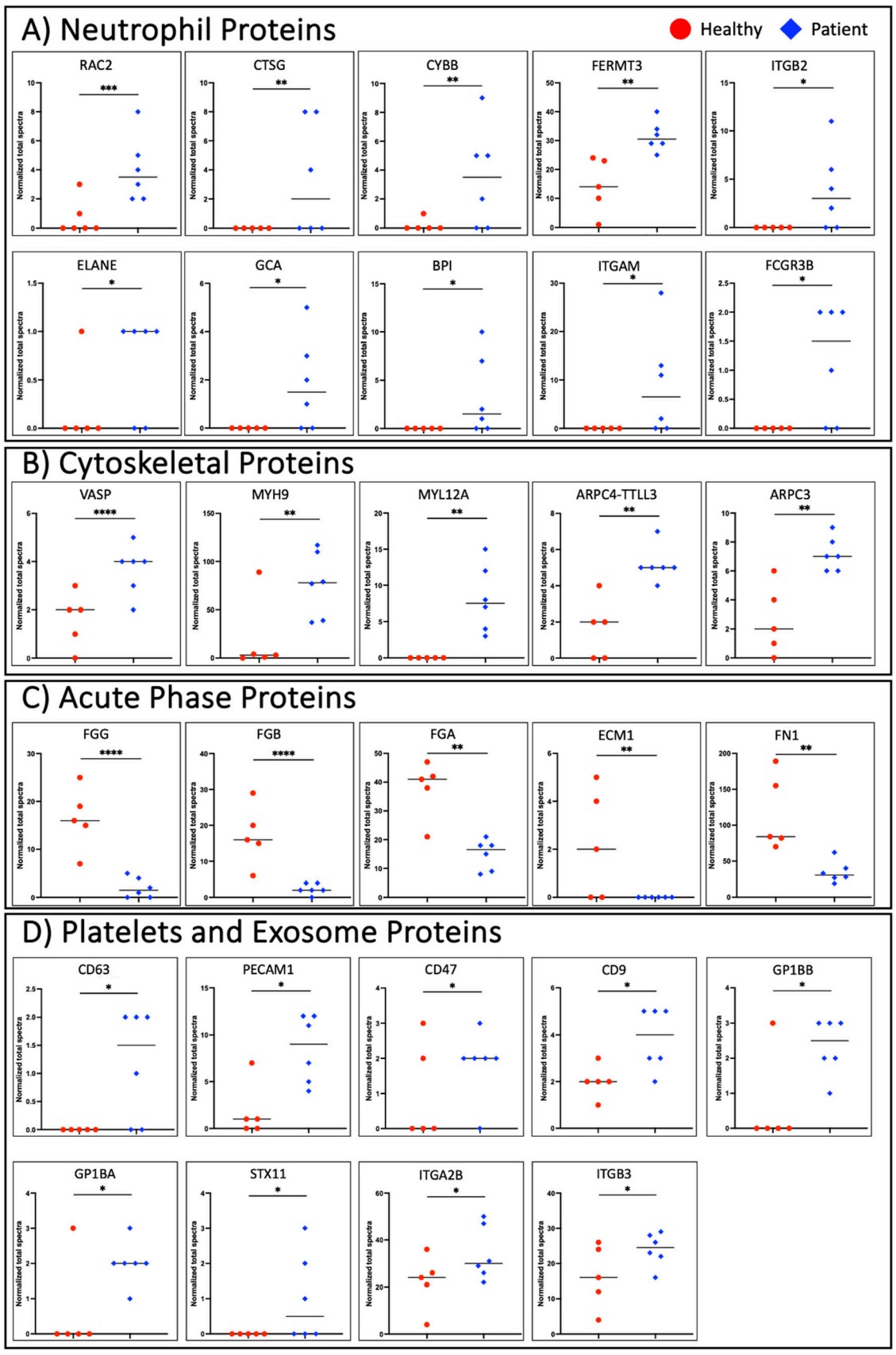

modify neutrophil function, thus imposing constraints on interpretation. It is important to note, however, that our previous murine studies had good correlation between the in vitro and in vivo observations of neutrophil NG uptake (Myerson et al, 2022). This study also includes a small number of samples because of the rare nature of STAT3 HIES disease. There were some findings that showed a trend but did not meet statistical significance so they were not discussed here. With more samples, this work will be able to be expanded on in future studies. We believe the results have two avenues of future study. The first next step from this work is investigation of the protein composition of healthy donor and STAT3 HIES patient serums. Here, we have postulated that the NG protein corona proteomic findings are because of serum composition differences between samples. Alternatively, it is possible that differences in STAT3 HIES patient proteins cause them to have a different affinity for opsonizing the NG particles. Proteomic analysis of raw serum will illuminate if our data are because of differences in the protein composition of the serums. The second next steps will be nanoparticle loading with cargo therapeutics and nanoparticle directed delivery of that cargo to alter and/or enhance neutrophil function.

In these studies, we found that human neutrophils are able to phagocytose NGs and have increased uptake when particles have previously been opsonized with serum. We have shown that although STAT3 HIES neutrophils phagocytose NGs in a similar amount to healthy controls under naive conditions, serum opsonization is not as effective to enhance particle uptake in this patient population. Furthermore, complement, specifically C3, was the most abundant protein bound to the NG in samples from both healthy controls and STAT3 HIES patients. However, there was significantly lower APP expression seen in protein samples from STAT3 HIES patients, consistent with their abnormalities of STAT3 signaling.

Our results suggest that although NANs may need further opsonization in this specific patient population, they have potential to be a viable drug delivery system for STAT3 HIES patients. With further study, NGs may have promise to improve the safety of delivery of antimicrobial agents for management of pneumonia and other severe infections.

## Materials and Methods

### Lysozyme-dextran nanogel synthesis

Lysozyme-dextran nanogels (NGs) were synthesized as previously described (Li et al, 2008; Ferrer et al, 2014; Myerson et al, 2018, 2022). FITC-dextran (Sigma-Aldrich) and lysozyme from hen egg white (Sigma-Aldrich) were dissolved in deionized and filtered water at a 1:1 or 2:1 mol:mol ratio. Then pH was adjusted to 7.1, and the solution was lyophilized. For Maillard reaction, the lyophilized product was heated for 18 h at 60°C, with 80% humidity maintained via saturated KBr solution in the heating vessel. Dextran-lysozyme conjugates were dissolved in deionized and filtered water to a concentration of 5 mg/ml. Solutions were stirred at 80°C for 30 min. Diameter of NGs was evaluated with dynamic light scattering (Malvern) after heat gelation. Particle suspensions were stored at 4°C.

### Human serum and neutrophils

Whole human blood for serum and neutrophil isolation was obtained with written informed consent on CHOP IRB protocol 20-017679 and through Material Transfer Agreement with the NIH. Whole human blood was collected in the Becton Dickinson red top Vacutainer. The Serum was prepared to manufacturer's recommendations. The blood was left undisturbed at 20°C for 30 min to clot. The clot was then removed by centrifugation at 1,500$g$ × 10 min. The serum was pipetted into Eppendorf tubes for immediate use or stored at –80°C. Human neutrophils were isolated from the whole human blood using the STEMCELL Technologies EasySep Human Neutrophil Isolation Kit (cat #17957) red blood cell lysis protocol exactly to manufacturer specifications.

### Nanogel preparation

FITC-labeled lysozyme-dextran NGs were synthesized as above. Stock NGs were brought to a concentration of 5 × 10$^{11}$ particles/ml. To serum-treat before neutrophil incubation, NGs were incubated in 50% matched-donor human serum in DMEM for 1 h at 37°C.

### Neutrophil-nanogel interaction

Neutrophils were isolated and prepared as above. 500 $\mu$l of neutrophils were incubated with 20 $\mu$l of either NGs in DMEM or NGs in 50% matched-donor human serum while rotating at 37°C for 15 min. Samples were then washed and pelleted 300$g$ × 6 min twice to remove unbound particles. After final wash, samples were resuspended in FACS buffer. For flow cytometry samples, they were stained with anti-CD66b antibody before analysis. For microscopy samples, they were suspended in solution with 2% paraformaldehyde for 30 min at room temperature, then pelleted and resuspended at a concentration of 1 × 10$^6$/ml.

---

**Figure 3. STAT3 HIES patient serum had a unique NG protein corona composition compared with the corona of healthy donor serum.**
NGs were incubated in either healthy donor or STAT3 HIES patient serum. They were then washed three times and prepared for mass spectrometry and protein profiling. Proteomic results were analyzed for peptides with a statistically significant differential between the healthy and patient samples. Shown are the results of the identified peptides that were present in at least 50% of the samples of either the healthy or patient group. In panel (A, B, and D), we see an overall increase in the amount of the proteins present in patient opsonized NGs compared with healthy donor. **(A)** The previous literature has shown that STAT3 is a negative regulator of neutrophil function, and here, there is an up-regulation of neutrophil proteins in the patient samples. **(B)** We hypothesize that the increase in cytoskeletal proteins seen in the protein corona created from patient serum may be related to the physiologic changes in response to dysfunctional neutrophils. In panel (D), we have shown increased exosome proteins, and cytoskeletal elements are found in exosome particles. We also propose that the increased platelet activity may increase NETosis, which cytoskeletal elements are included in the neutrophil extracellular traps. **(C)** STAT3 is a fundamental mediator of the induction of liver acute-phase genes. There is a significantly lower amount of each acute-phase protein in the patient samples as compared with the healthy. **(D)** Platelets and their exosomes enhance neutrophil response, including neutrophil recruitment to areas of inflammation and NETosis. In the patients who have dysfunctional neutrophils, we see an increase in the amount of platelet and exosome proteins identified in the serum protein corona of opsonized NGs.

## Sample preparation and confocal microscopy

90 $\mu$l of paraformaldehyde fixed cells was incubated with 5 × $10^{-7}$ M DAPI and 1:100 Vybrant DiD Cell-Labeling Solution (Invitrogen) at 37°C for 20 min. Samples were centrifuged at 2,000$g$ × 1 min. They were washed with 1 ml PBS and spun at 2,000$g$ × 3 min twice. Finally, they were resuspended in 80 $\mu$l PBS and then dropped on cavity slides (Eisco), covered with coverslips (Thermo Fisher Scientific), and analyzed with the Leica TCS SP8 Laser confocal microscope. Visualization of neutrophils was performed with water immersion objective HC PL APO CS2 40x/1.10. Images were obtained in the sequential scanning mode using Diode 405, OPSL 488, OPSL 552, and Diode 638 lasers. We used scan speed 200 Hz and pixel size 0.223 $\mu$m for flat images. Z-stacks were obtained with scan speed 600 Hz, XY pixel size 0.223 $\mu$m, and voxel size 0.424 $\mu$m. Images and Z-stacks were processed with LASX (Leica microsystems). Images and Z-stacks were converted to TIFF images for analysis in ImageJ (FIJI distribution 2.1.0/1.53q). Processing and analysis, including image thresholding, fluorescence colocalization, and per-cell analysis of fluorescence signals, employed custom ImageJ macros, code for which is provided in full in the supplement. For per-cell analyses, regions of interest were drawn manually around each imaged cell, using images of DiD membrane stain to define the edges of individual cells.

## Proteomics analysis

### In solution digestion
Sample was solubilized and digested with the iST kit (Kulak et al, 2017) (PreOmics GmbH) per manufacturers protocol. Briefly, the resulting pellet was solubilized, reduced, and alykylated by addition of SDC buffer containing TCEP and 2-chloroacetamide then heated to 95°C for 10 min. Proteins were enzymatically hydrolyzed for 1.5 h at 37°C by addition of LysC and trypsin. The resulting peptides were desalted, dried by vacuum centrifugation, and reconstituted in 0.1% TFA containing iRT peptides (Biognosys).

## Mass spectrometry analysis

Samples were block randomized and analyzed on a QExactive HF mass spectrometer (Thermo Fisher Scientific) coupled with an Ultimate 3000 nano UPLC System and EasySpray source. Peptides were separated by reverse-phase (RP) HPLC on Easy-Spray RSLC C18 2 $\mu$m 75 $\mu$m id × 50 cm column at 50°C. Mobile phase A consisted of 0.1% formic acid and mobile phase B of 0.1% formic acid/ acetonitrile. Peptides were eluted into the mass spectrometer at 300 nl/min with each RP-LC run comprising a 95-min gradient from 1 to 3% B in 5 min, 3–45% B, in 90 min. The mass spectrometer was set to repetitively scan m/z from 300 to 1800 (R = 120,000) followed by data-dependent MS/MS scans on the 20 most abundant ions, dynamic exclusion with a repeat count of 1, repeat duration of 15 s, (R = 45,000), and an nce of 27. FTMS full scan AGC target value was 5 × $10^5$, whereas MSn AGC was 1 × $10^5$, respectively. MSn injection time was 120 ms; microscans were set at one. Rejection of unassigned, 1, 6–8, and >8 charge states was set.

## Data analysis

MS/MS raw files for the DDA were searched against a reference human protein sequence database including reviewed isoforms from the Uniprot (19 October 2020), using MaxQuant version 2.0.1.0 (Tyanova et al, 2016). Trypsin was specified as enzyme with two possible missed cleavages. Carbamidomethyl of cysteine was specified as fixed modification and protein N-terminal acetylation, oxidation of methionine, and deamidation of asparagine and glutamine were considered variable modifications. The MS/MS tolerance FTMS was set to 20 ppm. False discovery rates were set to 1. The rest of search parameters were set to the default values.

## System suitability and quality control

The suitability of Q Exactive HF instrument was monitored using QuiC software (Biognosys) for the analysis of the spiked-in iRT peptides. Meanwhile, as a measure for quality control, we injected standard *E. coli* protein digest before and after the sample and collected the data in the Data Dependent Acquisition (DDA) mode. The collected DDA data were analyzed in MaxQuant, and the output was subsequently visualized using the PTXQC (Bielow et al, 2016) package to track the quality of the instrumentation.

## MS data processing and bioinformatics analysis

The raw files for DDA analysis were processed using MaxQuant version 1.6.14.0, with the reference human proteome sourced from UniProt (42,247 reviewed canonical and isoform proteins). Default MaxQuant settings were employed for peptide and protein quantification, with fixed modification for cysteine carbamidomethylation and variable modifications for methionine oxidation and protein N-terminal Acetylation. Filtering of data was performed at a false discovery rate of 1% at precursor, peptide, and protein levels. Subsequently, protein MS1 iBAQ intensity values were used for downstream bioinformatics analysis.

An in-house R package was applied for proteomics data processing and statistical analysis. The iBAQ values underwent $\log_2$ transformation and were normalized by subtracting the median value for each sample. Data filtering ensured the presence of at least three valid values for a protein in one cohort. To compare proteomics data between groups, Limma $t$ tests were employed to identify differentially abundant proteins. Volcano plots were generated to visualize the affected proteins when comparing different groups. Lists of differentially abundant proteins were then sorted based on a threshold of $P$-value < 0.05 and $|\log_2 FC| > 1$, resulting in a prioritized list for pathway enrichment analysis.

## Statistical analysis

Error bars indicate the SEM throughout. Significance tests are described in captions. Statistical power was determined for statements of statistical significance.

# Supplementary Information

# Acknowledgements

This research was supported by a grant from the Job Research Foundation (VP Krymskaya/KM Rubey). Authors of this study were also supported by grants from the National Institutes of Health F32 HL151026 and L40 HL159788 (KM Rubey); F32 HL162425, 5T32 HL007586-35, and KL2TR001879 (SM Lin); R01HL151467, R01 HL158737, R01 HL141462, R41 HL156767, U01 HL131022, and DOD TSCRP W81XWH2210503 (VP Krymskaya). The authors are grateful to Dr. George Scott Worthen for helpful and insightful discussions throughout this work and to Jacob Myerson for generously providing the nanogels used for these experiments. The authors declare no competing financial interests.

## Author Contributions

KM Rubey: conceptualization, resources, data curation, formal analysis, funding acquisition, investigation, methodology, and writing—original draft, review, and editing.

A Freeman: resources and writing—original draft.

AR Mukhitov: data curation, software, formal analysis, visualization, and writing—original draft, review, and editing.

AJ Paris: resources, funding acquisition, investigation, and methodology.

SM Lin: resources, methodology, and writing—original draft.

R Rue: resources, data curation, investigation, and writing—original draft.

H Fazelinia: data curation, software, formal analysis, and validation.

LA Spruce: data curation, software, and validation.

J Roof: data curation, software, and validation.

JS Brenner: conceptualization and methodology.

J Heimall: conceptualization, resources, formal analysis, funding acquisition, and writing—original draft, review, and editing.

VP Krymskaya: conceptualization, resources, data curation, formal analysis, funding acquisition, investigation, methodology, and writing—original draft, review, and editing.

## Conflict of Interest Statement

The authors declare that they have no conflict of interest.

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
