## [Reviewer comments · Life Science Alliance]

Life Science Alliance

Neutrophil-Avid Nanocarrier Uptake by STAT3 Dominant-Negative Hyper IgE Syndrome Patient Neutrophils

Kathryn Rubey, Alexandra Freeman, Alexander Mukhitov, Andrew Paris, Susan Lin, Ryan Rue, Hossein Fazelinia, Lynn Spruce, Jennifer Roof, Jacob Brenner, Jennifer Heimall, and Vera Krymskaya

DOI: <https://doi.org/10.26508/lsa.202402618>

Corresponding author(s): Kathryn Rubey, Children's Hospital of Philadelphia

Review Timeline:	Submission Date:	2024-01-24
	Editorial Decision:	2024-05-06
	Revision Received:	2024-07-24
	Editorial Decision:	2024-07-25
	Revision Received:	2024-08-02
	Accepted:	2024-08-05

Transaction Report:

May 6, 2024

Re: Life Science Alliance manuscript #LSA-2024-02618-T

Dr. Kathryn M Rubey
Children's Hospital of Philadelphia
Philadelphia

Dear Dr. Rubey,

Thank you for submitting your manuscript entitled "Uptake of Neutrophil-Avid Nanocarriers in Neutrophils from Patients with STAT3 Dominant Negative (DN) Hyper IgE Syndrome" to Life Science Alliance. The manuscript was assessed by expert reviewers, whose comments are appended to this letter. We invite you to submit a revised manuscript addressing the Reviewer comments.

Thank you for this interesting contribution to Life Science Alliance. We are looking forward to receiving your revised manuscript.

Sincerely,

B. MANUSCRIPT ORGANIZATION AND FORMATTING:

Reviewer #1 (Comments to the Authors (Required)):

Rubey et al present an interesting study where they augment the mechanistic alterations in neutrophils of patients suffering from STAT3-dominant negative Hyper IgE syndrome, a rare disease. The Authors have already reported defects in neutrophil functions in these patients, and also provided some mechanistic insights through animal models. Here, the manuscript focuses on the ability of patient neutrophils to uptake nanogels (NGs), as a founding stone on future delivery of a therapeutic treatment. The Authors have investigated the opsonisation of the NGs and studied differences between healthy versus patient serum samples in this process. Moreover, they have reported on the specific proteins that are present in the opsonisation process and compared between the two groups. The Authors have indicated the limitations of the study, the main one, in my view, is the small sample number. I have the following constructive comments:

1. While I understand the use of serum to opsonise the NGs, I think some of the speculations are too far fetched. As an example, Page 5, Lines 171-175: the difference may not reflect the actual in vivo situation, since serum generation provokes a massive activation of platelets and indeed leukocytes. This should be taken into account. In other words, while serum is required for opsonisation of the NGs, we cannot extrapolate differences to real-life differences in the blood of these patients. To this end, one should study plasma or indeed whole blood. Therefore, exosome numbers can be irrelevant as there must have been a massive release of them during the serum formation, as platelets and cells aggregates and are also activated by the coagulation cascade. The Authors could make some hypothesis on neutrophil-platelet interaction, as an example, if they analysed whole blood samples. Or at least have quantified not only neutrophils but also platelets in the blood of the different donors.

I think some of conclusions and hypotheses may be tempered by the fact that serum was used.

2. Page 4, Line 145. Is it surprising that NGs were associated with proteins that favour phagocytosis?

3. Page 4, Lines 153-155. I am not sure I can see why presence of higher protein amounts may be related to deficient neutrophil chemotaxis when cells were taken from the patients.

4. Page 5, Lines 186-188. The Authors can easily measure NETs and extracellular vesicles in the patient plasma samples, if available. Rather than offering a hypothesis only.

In addition:

5. Line 61, its and not it's

6. Line 137. I could not find Table 2.

7. Line 130. I think these are Figure 2C/2D.

8. Line 149. Is this meant to be elastane or elastase?

Reviewer #2 (Comments to the Authors (Required)):

Title- Uptake of Neutrophil-Avid Nanocarriers in Neutrophils from Patients with STAT3 Dominant Negative (DN) Hyper IgE Syndrome

Summary-

In this manuscript, Rubey et al demonstrated usefulness of lysozyme-dextran coated nanogel (NG) in both unopsonized and donor-matched serum opsonized healthy and patients with STAT3 HIES. Followed by neutrophile uptake assay of these NGs, authors also analyzed the protein components present in the protein corona surrounding the surface of NG particles opsonized with either healthy serum or HIES patient serum through mass spectrometry. These validation study can be highly useful as these NGs can be potential vehicle for delivering drugs that specifically target in patients like HIES with unusually low neutrophiles. However, authors require to address these following comments before acceptance of the

manuscript-

1. Authors presented the MFI in figure 1 but they need to show the gating strategy and main population flow cytometry diagrams.
2. It is not convincing from the current microscopy images that neutrophils uptake NG due to poor resolution, so authors are needed to provide more high-resolution images of neutrophil uptake from both healthy and HIES donors. This will strongly support their current explaining of obtaining less MFI in HIES neutrophils. Authors also need to provide secondary antibody control as the NG particles are FITC labelled which have a chance of autofluorescence.
3. Authors need to justify that in serum of healthy and HIES patient there is significant difference in- neutrophil proteins, cytoskeletal proteins, APPs and platelet exosome proteins to justify the differential NG protein corona.

Response to Reviewers

We appreciate the positive comments from the Reviewers, who described our study as possibly “highly useful as these NGs can be potential vehicle for delivering drugs that specifically target in patients like HIES with unusually low neutrophils.” We also thank the Reviewers for their constructive feedback, which we are confident has strengthened the manuscript. We have addressed the comments by editing the manuscript text and clarifying our conclusions. Below is a point-by-point response to all the Reviewer comments.

Reviewer #1

1. While I understand the use of serum to opsonise the NGs, I think some of the speculations are too far fetched. As an example, Page 5, Lines 171-175: the difference may not reflect the actual in vivo situation, since serum generation provokes a massive activation of platelets and indeed leukocytes. This should be taken into account. In other words, while serum is required for opsonisation of the NGs, we cannot extrapolate differences to real-life differences in the blood of these patients. To this end, one should study plasma or indeed whole blood. Therefore, exosome numbers can be irrelevant as there must have been a massive release of them during the serum formation, as platelets and cells aggregates and are also activated by the coagulation cascade. The Authors could make some hypothesis on neutrophil-platelet interaction, as an example, if they analysed whole blood samples. Or at least have quantified not only neutrophils but also platelets in the blood of the different donors. I think some of conclusions and hypotheses may be tempered by the fact that serum was used.

The use of serum for these in vitro assays is currently the best available model that we have given that we need to be able to recover the NGs after opsonization by centrifugation for incubation with neutrophils. Whole blood cannot be used as the centrifugation speeds need to pellet the NGs will also pellet the blood cells. Likewise, plasma is problematic because the anticoagulants available for use all have properties that will alter the nanoparticle's properties, and therefore, will not recapitulate the in vivo environment. One example is heparin's positive charge that alters nanoparticle function.

While we appreciate that these results may not perfectly align with an in vivo system, we would like to highlight that both patient and healthy serum was processed and prepared by the same protocol, and thus, the statistically significance differences in these proteins cannot be explained away by the serum preparation alone. We have presented a hypothesis that this may be related to the neutrophil-platelet interaction. Future studies will test this hypothesis further and include things like quantification of platelets in both healthy and patient blood samples.

2. Page 4, Line 145. Is it surprising that NGs were associated with proteins that favour phagocytosis?

We do not think that this is a surprising finding based on the in vivo findings, and was the basis of developing our hypotheses. However, this neutrophil tropism is not seen in

other nanoparticles, and our NGs have no specific neutrophil-targeting moieties, so we find this to be an interesting finding for these specific particles.

3. Page 4, Lines 153-155. I am not sure I can see why presence of higher protein amounts may be related to deficient neutrophil chemotaxis when cells were taken from the patients.

We developed this hypothesis as an explanation for the higher amount of specific neutrophil functional proteins found in patient serum samples, so the text has been edited to clearly communicate that this is just a hypothesis at this time. We propose that the deficient neutrophil chemotaxis causes difficulty for neutrophils homing to the location where they are needed to work, which ends up increasing recruitment signaling and upregulates the functional proteins that the patient neutrophils are able to make. We believe that upregulation in proteinases may be destructive of the extracellular matrix and of connective tissues, leading to patients also increasing production of proteinase inhibitors to attempt to reach homeostasis. This hypothesis will need to be further evaluated for future papers.

4. Page 5, Lines 186-188. The Authors can easily measure NETs and extracellular vesicles in the patient plasma samples, if available. Rather than offering a hypothesis only.

NETosis and extracellular vesicles are outside the scope of this lab's expertise. The NET and extracellular vesicle assays will be a focus of future work through collaboration with local experts.

5. Line 61, its and not it's

Text has been edited.

6. Line 137. I could not find Table 2.

Table 2 should be attached in these resubmission files.

7. Line 130. I think these are Figure 2C/2D.

Text has been edited to reference the proper figure.

8. Line 149. Is this meant to be elastane or elastase?

Typo has been edited in text.

Reviewer #2

1. Authors presented the MFI in figure 1 but they need to show the gating strategy and main population flow cytometry diagrams.

Figure 1 has been updated and includes the gating parameters used.

2. It is not convincing from the current microscopy images that neutrophils uptake NG due to poor resolution, so authors are needed to provide more high-resolution images of neutrophil uptake from both healthy and HIES donors. This will strongly support their current explaining of obtaining less MFI in HIES neutrophils. Authors also need to provide secondary antibody control as the NG particles are FITC labelled which have a chance of autofluorescence.

The microscopy has been updated to the highest resolution image possible. We have also included an orthogonal projection to give a better special image of the NGs within the neutrophils. Our protocol uses nanogels synthesized from FITC-dextran (Page 6, lines 209-211), meaning the particles themselves are fluorescent without any tagging. As there is no antibody used to label the NG particles, there is not an antibody control indicated for these experiments.

3. Authors need to justify that in serum of healthy and HIES patient there is significant difference in- neutrophil proteins, cytoskeletal proteins, APPs and platelet proteins to justify the differential NG protein corona.

This was also a question that we had upon received our protein corona results. We discussed ways we could investigate the serum itself to compare to the protein corona formed on NGs with our proteomics core experts, and unfortunately, current technology does not exist to answer them. Due to the large amounts of albumin found in serum, the current mass spectrometry analysis capabilities are unable to quantify most of the less abundant proteins, which are the ones found in the NG protein corona. We hope to pursue this question as technology advances in future studies.

July 25, 2024

RE: Life Science Alliance Manuscript #LSA-2024-02618-TR

Dr. Kathryn M Rubey
Children's Hospital of Philadelphia
3401 Civic Center Blvd
Philadelphia 19104

Dear Dr. Rubey,

Thank you for submitting your revised manuscript entitled "Neutrophil-Avid Nanocarrier Uptake by STAT3 Dominant-Negative Hyper IgE Syndrome Patient Neutrophils". We would be happy to publish your paper in Life Science Alliance pending final revisions necessary to meet our formatting guidelines.

- please be sure that the authorship listing and order is correct
- please add a Summary Blurb/Alternate Abstract to our system
- please add the Twitter handle of your host institute/organization as well as your own or/and one of the authors in our system
- please add the Abstract to the manuscript file
- please incorporate any points from the Conclusion section into the Discussion; we only allow a Discussion section
- please consult our manuscript preparation guidelines <https://www.life-science-alliance.org/manuscript-prep> and make sure your manuscript sections are in the correct order
- please add your main, supplementary figure, and table legends to the main manuscript text after the references section
- please use the [10 author names et al.] format in your references (i.e., limit the author names to the first 10)
- please add an Author Contributions section to your main manuscript text
- please add a Conflict of Interest statement to your main manuscript text
- please add callouts for Figures 1C-E and S1A-B to your main manuscript text
- please indicate that written informed consent was obtained from the patients

FIGURE CHECKS

- please add a scale bar to Figure 1E

LSA now encourages authors to provide a 30-60 second video where the study is briefly explained. We will use these videos on social media to promote the published paper and the presenting author (for examples, see <https://docs.google.com/document/d/1-UWCfbE4pGcDdcgzcmiuJl2XMBJnxKYeqRvLLrLS08s/edit?usp=sharing>). Corresponding or first-authors are welcome to submit the video. Please submit only one video per manuscript. The video can be emailed to contact@life-science-alliance.org

A. FINAL FILES:

B. MANUSCRIPT ORGANIZATION AND FORMATTING:

Thank you for your attention to these final processing requirements. Please revise and format the manuscript and upload materials within 5 days.

Sincerely,

August 5, 2024

RE: Life Science Alliance Manuscript #LSA-2024-02618-TRR

Dr. Kathryn M Rubey
Children's Hospital of Philadelphia
3401 Civic Center Blvd
Philadelphia 19104

Dear Dr. Rubey,

Thank you for submitting your Research Article entitled "Neutrophil-Avid Nanocarrier Uptake by STAT3 Dominant-Negative Hyper IgE Syndrome Patient Neutrophils". It is a pleasure to let you know that your manuscript is now accepted for publication in Life Science Alliance. Congratulations on this interesting work.

DISTRIBUTION OF MATERIALS:

Again, congratulations on a very nice paper. I hope you found the review process to be constructive and are pleased with how the manuscript was handled editorially. We look forward to future exciting submissions from your lab.

Sincerely,
